# Immunomodulatory Properties of Host Defence Peptides in Skin Wound Healing

**DOI:** 10.3390/biom11070952

**Published:** 2021-06-28

**Authors:** Marija Petkovic, Michelle Vang Mouritzen, Biljana Mojsoska, Håvard Jenssen

**Affiliations:** Department of Science and Environment, Roskilde University, 4000 Roskilde, Denmark; marijap@ruc.dk (M.P.); mvang@ruc.dk (M.V.M.); biljana@ruc.dk (B.M.)

**Keywords:** antimicrobial peptides, host defence peptides, chronic wounds, skin wound healing, inflammation, skin immune response

## Abstract

Cutaneous wound healing is a vital biological process that aids skin regeneration upon injury. Wound healing failure results from persistent inflammatory conditions observed in diabetes, or autoimmune diseases like psoriasis. Chronic wounds are incurable due to factors like poor oxygenation, aberrant function of peripheral sensory nervature, inadequate nutrients and blood tissue supply. The most significant hallmark of chronic wounds is heavily aberrant immune skin function. The immune response in humans relies on a large network of signalling molecules and their interactions. Research studies have reported on the dual role of host defence peptides (HDPs), which are also often called antimicrobial peptides (AMPs). Their duality reflects their potential for acting as antibacterial peptides, and as immunodulators that assist in modulating several biological signalling pathways related to processes such as wound healing, autoimmune disease, and others. HDPs may differentially control gene regulation and alter the behaviour of epithelial and immune cells, resulting in modulation of immune responses. In this review, we shed light on the understanding and most recent advances related to molecular mechanisms and immune modulatory features of host defence peptides in human skin wound healing. Understanding their functional role in skin immunity may further inspire topical treatments for chronic wounds.

## 1. Introduction

Acute wounds in diabetic patients can adopt and portray the chronicity of the non-healing wounds due to bottom-line complications like the duration of diabetes or vascular disease paired with peripheral neuropathies [1]. Wounds that do not heal naturally within three months are defined as chronic wounds, and often require treatment to heal. Wounds that have healing difficulties are subcategorised into four aetiology categories: venous, pressure, diabetic and arterial insufficient ulcers. Non-healing wounds share a lack of oxygen and nutrient supply and microbial contagion, delaying the wound from healing [2]. Diabetic foot ulcerations occur in about 20% of the diabetic population, being prevalent among the chronic wound aetiologies, like venous and pressure ulcers [3].

Other inflammatory skin conditions like psoriasis [4,5] and atopic dermatitis [6] are also characterised by dysregulation of the immune response, attacking healthy skin cells.

Host defence peptides are polycationic peptides exhibiting various antimicrobial activities or prompting the host’s immune responses. These agents are naturally produced by a wide variety of species, ranging from marine organisms to humans. Recent studies propose the dual importance of host defence peptides (HDPs) in the different phases of wound healing [7,8]. As the first line of defence against pathogenic bacterial infection, HDPs are a critical element in preventing biofilm-associated infections [9,10]. A decline in sufficiently effective antibiotic treatments due to growing problems with antibiotic resistance may be ameliorated with alternatives to conventional antibiotics like peptides with antimicrobial properties [11]. Aside from bacterial pathogen inactivation by disrupting their cell membranes, host defence peptides also have immunomodulatory properties, due to their ability to stimulate the cross-talk between immune cells promoting cutaneous wound healing in a healthier manner [12].

Different HDPs share some similarities like the number of amino acid residues being between 10 and 60, a cationic charge of 2–9 and depending on the sequence length, HDPs are classified as long (50–100 amino acids), intermediate (25–50 amino acids) or short (9–24 amino acids) [13,14]. The diverse structure yet positive net charge that they have in common is an important prerequisite for the design of the more stable, synthetic analogues that interact with anionic prokaryotic membranes: lipopolysaccharides (LPS; in Gram-negative bacteria) and teichoic acids (in Gram-positive bacteria) [11]. Host defence peptides and their synthetic analogues called peptidomimetics, which contain sequences built by natural and unnatural amino acids [15]. These building blocks determine the signature physiochemical properties, which are the charge (neutral or positive) and the hydrophobicity/hydrophilicity. These in turn contribute to the other level of structural complexity that plays a significant role in the activity: the secondary structures of HDPs [16,17]. Aside from their known potential for evading infections, an increasing body of evidence has demonstrated that HDPs are able to exert intracellular inhibitory activities as the primary or supportive mechanisms to achieve efficient killing [18]. The latter activity is referred to as immunomodulatory activity, and multiple studies have presented evidence on this important role in innate and adaptive immune response [19]. Most of the HDP derivatives include a combination of microbicidal action and immunomodulatory functions [20]. The challenges associated with creating a non-immunogenic peptide without the potentially adverse effects observed in natural HDPs may be defeated by the modification of internal sequences or single amino acid substitutions [21,22]. An example of such peptides is innate defence regulator peptides (IDR peptides) [23,24]. Specific signature structure properties that allow synthetic peptides to exert immunomodulatory properties are yet not well defined in the literature and are quite diverse.

## 2. Host Defence Peptides in Wound Healing

Cationic peptides play a primarily role in maintaining the skin barrier’s integrity and cutaneous tissue restoration during injury [25]. Due to expanding the effects of antimicrobial peptides on bacteria, HDP have been assigned the fundamental biological role in innate immunity [26]. Due to their lack of adaptive immune systems, arthropods and plants rely on their HDPs’ primary defence response [27,28]. In higher eukaryotes, the levels of the host’s defence peptides patrolling through signalling networks of the immune response are significantly more abundant, as their multifaceted role is more complex [29].

Skin resident HDPs are crucial participants of each step of the wound healing process: inflammation (neutrophil and macrophage infiltration), wound site regeneration (angiogenesis and re-epithelialization) and remodelling (restoration of tensile strength) [25,30].

In humans, two main classes of host defence peptides have been identified: defensins and cathelicidins (Figure 1). However, there are also a variety of other small peptides expressed by epithelial cells like Substance P [31], neurotensin [32], granulysin [33], calprotectin [34], adrenomedullin [35], MRP8/MRP-14 [36] and RNase A superfamily [37], which are also important wound inflammatory biomarkers.

### 2.1. Defensins

Peptides from the defensin family are small peptides that are widely distributed across species, including humans. Human defensins (HDs) are produced in leukocytes and are also secreted by different epithelial cells and mucosal tissues [38]. The mature defensins are described as short (28–42 amino acids length), cationic (net charge +1 to +11), amphipathic peptides with a highly conserved tertiary structure of a triple antiparallel β-sheet fold arrangement accommodating the six cysteine residues connected with three disulphide bridges [39].

Moreover, depending on the size, location, spatial conformation, and spot where the cysteines lay within the peptide chains, they are categorised as α-defensins, β-defensins and θ-defensins [39,40]. In humans, only α and β-defensins are present, while structurally different, cyclic θ-defensins have been identified in rhesus macaques [39,41].

Additionally, the HDs have antimicrobial activity against various strains of Gram-positive and negative bacteria [42], fungi [43] and viruses such as the herpes simplex virus [44].

In humans, there are 6 α-defensins: HNPs1–4 and human α-defensins 5 and 6 (HD5 and HD6) [42,45]. HNPs 1–3 (human neutrophil peptides) differ by single amino acid substitutions and are predominantly produced by neutrophils.

Moreover, β-defensins abundance is tightly governed by transcriptional controls assigned to epithelial and epidermal cells [46]. One representative of the β-defensin family such as human β-defensin 1 (HBD1) is continuously transcribed in skin cells, while the transcription of human β-defensin 2 (HBD2) and human β-defensin 3 (HBD3) as the others is triggered in response to a microbial or pro-inflammatory cytokines presence [47].

### 2.2. Cathelicidins

Endogenous cathelicidins are stored at high concentrations as inactive precursors in granules of mammalian neutrophils and mast cells. In contrast to defensins, cathelicidins are predominantly α-helical, amphipathic, cationic (possessing a net positive charge of 2–9) and consist of 23–37 amino acids [48].

Cathelicidins are named after the conserved cathelin-like domain, and are produced as protein precursors, containing a conserved amino-terminal (N-terminal) signal peptide domain, an antimicrobial C-terminal domain mature peptide and cathelin-like domain [49].

The active form of peptide is formed once the cathelin domain is cleaved by serine proteases upon neutrophils degranulation and secretion of peptides [50].

The gene for hCAP-18, named *CAMP*, encodes the human cathelicidin. LL-37, the only member of the cathelicidin family identified so far in humans, is present in the secondary (specific) granules of neutrophils, macrophages, and epithelial cells [51]. Human cathelicidin hCAP-18, is processed by proteinases into several extracellular cleavage products, with LL-37 being one of them [52]. Cathelicidins play an important regulatory role in the inflammatory response [53]. Many studies support immunomodulatory actions over the antimicrobial actions of cathelicidins [26,48]. However, the constitutive generation of the cathelicidins across the various species provides them with strong antimicrobial protection from bacteria [54], viruses [55] or fungi [56].

## 3. Wound Healing Phases

The skin is a multifunctional organ whose outermost position provides protective a barrier [57]. In addition to providing mechanical protection and support to internal organs, the subtle cutaneous microenvironment has the capacity to actively mediate immune response [58]. The intact skin is composed of three layers: epidermis, dermis and hypodermis, with the epidermis being the outermost layer and the hypodermis the innermost layer.

When a wound occurs in the skin, healing is essential for the skin to restore the integrity. The wound healing is a perplexed process consisting of three sophistically coordinated phases: inflammation, proliferation, and remodelling [59]. However, this classification is arbitrary, as those phases are overlapping, and even distant areas of wounds can be in different phases of healing [60] (Figure 2).

### 3.1. Inflammation

The inflammatory phase begins with hemostasis. Hemostasis involves various protease cascades leading to the formation of a fibrin cloth to prevent blood loss and seal the wound [61]. The inflammatory phase is characterised by infiltration of immune cells, such as neutrophils, macrophages, and lymphocytes, which aim to eliminate pathogens and cellular debris from the wound site [62].

Neutrophils as the first responders engulf the pro-inflammatory cytokines, with DNA, RNA and other cellular components often referred to as damage-associated molecule patterns [63]. Phagocytic products trigger the release of cytokines, growth factors, production of reactive oxygen species (ROS) and proteolytic enzymes, which in turn attract more immune cells to the wounded site [64]. Another highly abundant subpopulation of immune cells are macrophages. During wound healing, two subgroups of macrophages populate the wound depending on the cytokine secretion profile: pro-inflammatory or M1 and anti-inflammatory or M2 [65]. The macrophage polarisation to either the pro- or anti-inflammatory phenotype is tightly controlled by signalling pathways, transcriptional and posttranscriptional regulatory networks [66].

### 3.2. Proliferation

The proliferative phase is characterised by re-epithelialization, angiogenesis and formation of granulation tissue that leads to closure of the epithelial layer, revascularization in the damaged area and tissue regeneration [67].

Epidermal growth factor (EGF) release initiates the re-epithelialization of the epidermis. This stimulates the keratinocytes, as the predominant cell type in the epidermis, to protrude, adhere, contract, and detach, migrating this way under the fibrin clot [68]. Angiogenesis is the formation of new blood capillaries existing from vessels. This is crucial, as healing requires energy for cell proliferation, migration, and production of collagen. Angiogenesis is stimulated by vascular endothelial growth factor (VEGF) [69]. The granulation tissue restoration formation is necessary for the connective tissue restoration, and it is performed by fibroblasts that synthesize the extracellular matrix (ECM) and collagen to strengthen the new tissue [70].

### 3.3. Remodelling

The remodelling phase can span for several years [71]. The remodelling or maturation phase involves degradation of the ECM and collagen III to collagen I replacement, resulting in increased tensile strength of the newly formed tissue [72,73]. The fibroblasts differentiation into myofibroblasts aids reduction of the wound size. Once the contraction of the wound is completed, the number of immune cells [74], the vessels [75] and the myofibroblasts undergo apoptosis [76]. The newly formed fibres and collagen structures are disorganised and can take years before they are properly reorganised to form fully healed tissue [77].

## 4. Cellular Regulation and Immunomodulatory Actions of Host Defence Peptides during Wound Healing

Factors affecting wound healing can be categorised into systemic and local factors. Systemic factors are present at all times in the individual, as seen in diseases such as diabetes, stress, obesity and age, while local factors include factors, such as infection, oxygenation and foreign bodies, which directly influence the wound locally [78]. The local factors lead to high levels of pro-inflammatory cytokines and ROS, impaired cell and protease function and a lack of growth factors [79]. HDPs have been demonstrated to influence many local factors, such as regulating expression of cytokines [80], chemokines [81], proteases [82], growth factors [83] and immune cell activity [84,85,86] (Table 1).

### 4.1. Host Defense Peptides Triggered by Inflammation

Many acute and chronic inflammatory disorders have been correlated with dysregulation of the natural HDP response [26,98]. Although the reciprocity of either deficiency or overproduction of HDPs in the presence of inflammation and thus balance between pro- or anti-inflammatory effects is not so easy to define directly, this can lead to a pathological inflammatory response [99,100]. Endogenous host defence peptides, stored intracellularly at high concentrations as inactive precursors in granules, are released locally at infection and inflammation sites, whereas the expression of others is induced in response to pathogen-associated molecules [101].

The role of the immune cells in the inflammatory phase is intended to eliminate intracellular pathogens, and this is achieved by producing host defence peptides, which in turn promote the robust recruitment of immune cells such as neutrophils, monocytes/macrophages, dendritic cells, and T cells [26]. Altered levels of either pro- [102,103] or anti-inflammatory [81,104,105] cytokines have been reported in inflammatory conditions. A persistent inflammatory environment has been demonstrated to affect not only the function of immune cells [106], but also the composition of the host defence peptides detected at the wound site [107] (Figure 3).

Defensins and cathelicidins family members have distinct roles in various disease states, and many studies support the potential synergistic roles of defensins and cathelicidins in cytokines chemoattraction in skin immune responses [91]. Tightly coordinated actions of hBD-2, -3 and -4 along with LL-37 induce secretion of IL-18 by keratinocytes through activation of the p38 and ERK1/2 MAPK pathways [91]. Under inflammatory skin conditions, high expression levels of keratinocyte hBD-2 induced by IL-17A and TNFα indicate that it may be considered as a marker for disease severity [108].

Inflammatory skin conditions like psoriasis are linked to elevated levels of both IL-1a and S100A7 (also known as psoriasin) and an increased signalling axis composed of NF-κB/p38MAPK/Caspase-1/IL-1a, which regulates S100A7 [93]. Moreover, the increased expression of LL-37 is associated with overproduction of IFNγ in psoriatic lesions, while hBD-2, hBD-3 and lysozyme can activate plasmacytoid dendritic cells (pDCs) in co-action with LL-37 [92]. LL-37 evokes pro-inflammatory chemotaxis through binding to the formyl peptide-like receptor-2 (Fpr-2), a cell sensing element for microbial products [88]. The human host defence peptide LL-37 binds to these G protein–coupled receptors and activates mucosal immune response through recruitment of leukocytes [88]. In contrast, LL-37 can mediate anti-inflammatory responses such as secretion of anti-inflammatory cytokines like IL-1RA [89]. Another prominent feature of the neutrophil released cathelicidins is enhanced SMAD2/3 and STAT3 phosphorylation in the presence of transforming growth factor-β (TGFβ), shifting the T-cells subset towards Th17 rather than the Th1 phenotype, resulting in Th17 but not Th1 cells protection from apoptosis [90].

Atopic dermatitis (eczema) is another pathological skin condition [109]. Unlike the psoriasis which is characterised by elevated levels of innate immunity peptides, in atopic dermatitis, host defence peptide levels are not abundant and together with skin dryness make affected areas prone to infections [110].

### 4.2. Host Defense Peptides Involved in Proliferative Phase of Wound Healing

It has been shown that LL-37 peptide induces growth factors, such as EGF and VEGF, but also binds their receptors [87,111]. Aberrant vascularization during wound restoration in mice lacking the CRAMP (the murine homologue of LL-37/hCAP-18), shows that the peptide can stimulate endothelial cells, increasing the proliferation rate and thus enhancing the dermal neovascularization [94].

Furthermore, high levels of hBD-2 and hBD-3 detected at wound sites promote keratinocyte migration and proliferation, indicating their involvement in the re-epithelialization of the healing epithelium [95,112]. A relatively recent study demonstrated that topical insulin delivery to the wound enhanced the levels of extracellular-signal regulated kinase (ERK) and protein kinase B (Akt) [113]. ERK is a part of a phosphorylation pathway that activates gene transcription leading to cell [114,115] and increased Akt stimulated angiogenesis through activation VEGF signalling [116,117] (Figure 4).

### 4.3. Host Defense Peptides in Tissue Remodelling Phase

The direction of macrophages polarisation into relevant phenotypes during different phases of wound healing (i.e., polarisation to a pro-inflammatory phenotype during the inflammatory phase and anti-inflammatory phenotype during the subsequent proliferation and remodelling phase) [118,119] is controlled by many factors, while different HDPs (like LL-37) promote polarisation to M1 macrophages [53]. Similarly, elevated levels of interleukin-6 (IL-6), IL-1β, tumour necrosis factor-α (TNFα) and chemokine CCL3 in RAW264.7 cells can be dampened by Cathelicidin-WA (CWA) [96]. CWA peptide suppresses the expression of TLR-4 and the phosphorylation of STAT1 and NF-κB, downregulating the activity of pro-inflammatory macrophages while stimulating the phosphorylation of STAT6 and activating *E. coli* K88-induced anti-inflammatory macrophages [96].

An aberrant response described as overgrowth of fibroblasts and endothelial cells may be due to high concentrations of psoriasin in wound fluid and granulation tissue [97]. Psoriasin stimulates the tissue remodelling by inhibiting the excessive production of collagen, fibronectin and TGFβ formation in fibroblasts. In keloid-derived fibroblasts, psoriasin production is decreased [97] (Figure 4).

## 5. Immunomodulatory Host Defence-Inspired Peptide Wound Treatments

Many host defence-inspired peptides are used as topical dermal treatments for wound healing [25], and the list of experimentally verified alternatives peptides is long [24,120,121,122,123,124]. One of the most widely used peptides for wounds with healing impairments and infections is LL-37 [125,126,127,128,129,130]. Transdermal delivery of peptides may be enhanced using novel 3D nanofiber scaffolds, to overcome previously reported poor cellular penetration [125]. The CO_2_ expanded nanofiber scaffolds can greatly promote cellular infiltration, neovascularization, and positive host response after subcutaneous implantation of coated LL-37 peptide [125]. Nanoparticles lipid carriers (NLCs) encapsulating LL-37 and administered through the topical route, have demonstrated to accelerated wound closure along with re-epithelization and decreased the inflammation in vitro and in vivo [126]. Moreover, the topical administration of LL-37 encapsulated in PLGA nanoparticles accelerated the wound closure along with re-epithelialization and improved the structure of the granulation tissue at the wound bed [127]. In addition to this, significantly up-regulated IL-6 and VEGF expression modulated the inflammatory wound response, leading to neovascularization improvements [127].

LL-37-conjugated gold nanoparticles exhibited enhanced in vivo wound healing activity compared to LL-37 alone by improving cell migration mediated by EGFR and ERK1/2 phosphorylation [131]. Wounds treated with LL-37-conjugated gold nanoparticles exhibited a better structure of collagen fibres IL-6 and VEGF [131]. Chronic diabetic wounds treated with antimicrobial peptide (LL-37) and fused with ultra-small gold nanoparticles (AuNPs) as a gene delivery system supported accelerated wound closure, as the wounds were permeated with newly formed blood vessels and the bacterial load was reduced [128]. In addition to, faster re-epithelization, granulation tissue improved, and VEGF expression increased [128]. LL-37 inspired wound drugs offer efficiency but require delivery stability through artificially synthesized carriers like calcium phosphate CaP nanoparticles [129]. Coating the LL-37 on CaP offers protection against enzymatic degradation, while the biological functionality and antimicrobial activity against both Gram-positive and Gram-negative bacteria remains intact [129].

Another wound healing therapeutics inspired by human α-defensin 5 (HD5) is nanodefensin (ND) with dual antimicrobial/immunomodulatory action [130]. The coating with upgraded pharmacological stability, a nanodefensin-encased hydrogel (NDEFgel) locally applied to the wounded surface, accelerated wound regeneration, and increased the expression of myofibroblasts and GTP-binding protein Rac1 [130].

Neuropeptides released from peripheral nerves like neurotensin display amelioration of wound healing impairments when loaded on collagen dressings by reducing the inflammation in diabetic mice wounds [132]. Exogenous delivery of the neurotensin does not translate into functional modifications on keratinocytes, particularly in terms of migration [117,133]. Another peptide with relevance to wound healing is Substance P, an undecapeptide (11 amino acids long) member of the tachykinin neuropeptide family [134]. Its effects are exerted via a high-affinity neurokinin-1 receptor (NK1R), and NK1RKO mice show higher Substance P expression [135]. The local treatment with Substance P or analogues has the potential not only to promote diabetic foot ulceration healing but also to modulate inflammation and macrophage phenotype [117,135]. An injectable Laponite nanodiscs-based hydrogel loaded with Substance P promoted wound healing by delivering the Substance P to a tissue-engineered skin model to stimulate the reepithelialization process [136].

## 6. Conclusions and Perspectives

HDPs are a vital component of the innate immune system of all eukaryotic organisms. In addition to their ability to kill microbial pathogens directly, HDPs can indirectly modulate the host defence systems. Dysregulation of their expression not only in the skin, but also in other body sites, can contribute to various pathological states. There are increasing efforts to further characterize peptide interactions with the immune system of various novel human HDPs as well as synthetic analogues and increase our knowledge of their role in wound healing. Their immunomodulation activity is promising for the development of effective drug adjuvants and antimicrobial therapeutics.

## Figures and Tables

**Figure 1 biomolecules-11-00952-f001:**
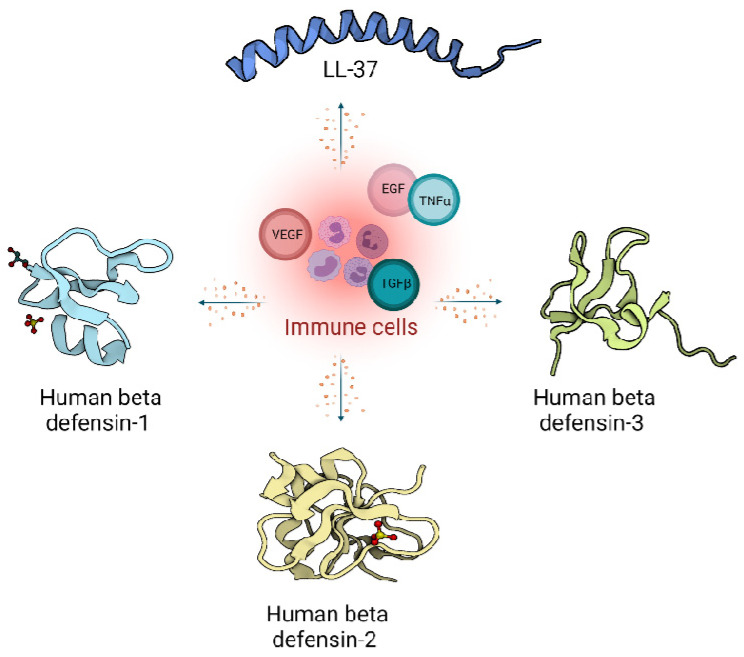
Host defence peptides expressed by skin cells are induced by immune response, and act via signalling proteins belonging to growth factors like Vascular Endothelial Growth Factor (VEGF), Transforming Growth Factor beta (TGFβ), Epidermal Growth Factor (EGF) and cytokines like Tumor Necrosis factor alpha (TNFα). Immune cells produce LL-37 from the cathelicidin family (Pdb code: 2K6O) [10] and defensin family: Human defensin 1 (Pdb code: 1IJU) [38], Human defensin 2 (Pdb code: 1FD4) [39], Human defensin 3 (Pdb code: 1KJ6) [40].

**Figure 2 biomolecules-11-00952-f002:**
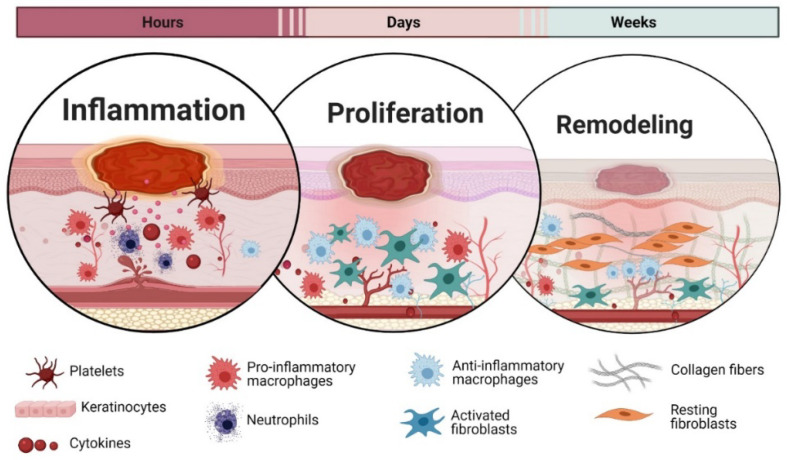
Wound healing phases. The timeline and overview for the healing phases also shows the involvement of the key cellular populations involved in the different phases.

**Figure 3 biomolecules-11-00952-f003:**
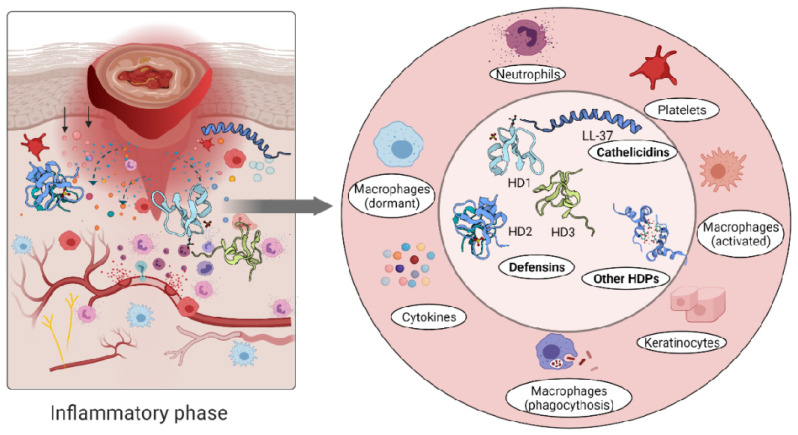
Immune response triggers skin cells to secrete host defence peptides. The secreted peptides are involved in antibacterial killing but also the recruitment of other immune cells to the wounded area.

**Figure 4 biomolecules-11-00952-f004:**
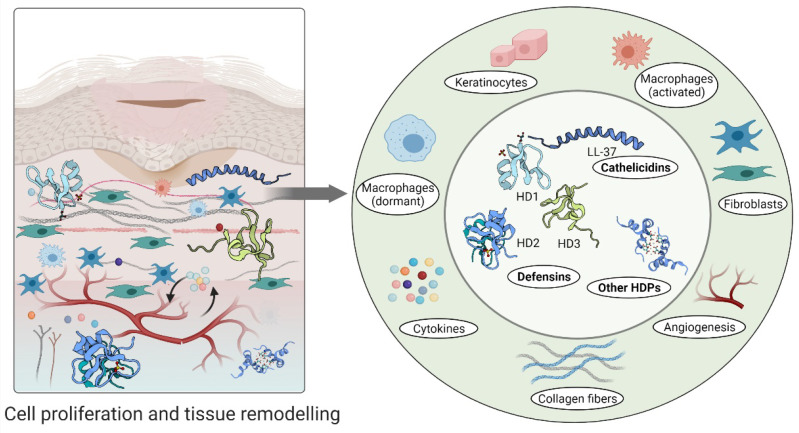
Host defence peptides involved in later stages of wound healing to assist skin regeneration.

**Table 1 biomolecules-11-00952-t001:** Host defence peptides immunomodulatory actions in wound healing phases.

Human Host Defence Peptide	Immunomodulatory Action	Reference
**Inflammatory phase**		
LL-37	Cell migration mediated by EGFR and ERK1/2	[87]
	Recruitment of leukocytes	[88]
	Secretion of anti-inflammatory cytokines like IL-1RA	[89]
	Enhanced SMAD2/3 and STAT3 phosphorylation	[90]
LL-37, hBD-2, hBD-3, hBD-4	Activation of the p38 and ERK1/2 MAPK pathways	[91]
hBD-2, hBD-3	Activate plasmacytoid dendritic cells (pDCs)	[92]
S100A7	Increased NFκB/p38MAPK/Caspase-1/IL-1a signalling	[93]
**Proliferative phase**		
LL-37	Induction of growth factors such as EGF and VEGF	[87]
	Neovascularization	[94]
hBD-2, hBD-3	Keratinocyte migration and proliferation	[95]
**Remodelling phase**		
LL-37	Polarisation to M1 macrophages	[53]
Cathelicidin-WA (CWA)	Suppressed phosphorylation of STAT1 and NF-κB, repressed phosphorylation of STAT6	[96]
S100A7	Aberrant response of fibroblasts and endothelial cells	[97]

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
