# Peer review of "Immunomodulatory Properties of Host Defence Peptides in Skin Wound Healing"

_biomolecules, 2021, doi:10.3390/biom11070952_

Round 1

Reviewer 1 Report

Reviewer report to the review paper entitled „Immunomodulatory properties of host defense peptides in skin wound healing“ by Marija Petkovic, Michelle V. Mouritzen, Biljana Mojsoska and Håvard Jenssen.

In this review paper the authors present current research on the involvement of host defence peptides in skin immunity with special focus on wound healing and in parts on inflammatory skin conditions like psoriasis and atopic dermatitis. The introduction lead to the topic of wounds with healing difficulties and a tendency to develop chronic wounds as well as inflammatory skin conditions. The authors selected defensins and cathelicidins and explain their modulatory action in the different phases of wound healing and in inflammatory skin conditions. Finally, the authors reviewed recent achievements on host defense- inspired wound treatments, i.e. as wound dressings or topical dermal applications.

The main immunomodulatory actions are in addition to the text also summarized in a table for better visualisation. Moreover, several figures illustrate very compact the processes described and cellular component involved. The order and selection of chapters is well chosen and follows a comprehensible logic.    

In chapter 2.1 and 2.2 the information given do not follow the same pattern. While for defensins detailed information on the structural arragement of the protein is given, this is not mentioned for cathelicidins. Moreover, the specific described antimicrobial activity should be added in the style of line 86-88 for cathelicidins.  

Both terms ‚antimicrobial peptide‘ and ‚host defence peptide‘ are used as synonyms. While only the term AMP was used in the introductory part, this changed to HDP in chapter 2. It would be good if the authors used one term consitently or added that they can be used synonymously. Possibly it is meant differently.

specific comments:

For chapter 4 the heading is missing (line 150) or wrongly numbered.  

Line 132-133 should read epidermis.

It seems a word is missing in the sentence beginning line 256. Please check.

Occasionally, some minor spelling mistakes occured throughout the manuscript, i.e. line 177 -179. Please recheck.  

The list of references is not consistently formatted in the same style. Please check especially ref.4, 5 and 19-21. In some references the doi is given plus page numbers and volume and in others it is not or only the doi.  

Author Response

Dear Reviewer

Thanks for a thorough and constructive review.

All points have been addressed and marked in the revised manuscript.

The points raised are marked in italic with a response following:

In chapter 2.1 and 2.2 the information given do not follow the same pattern. While for defensins detailed information on the structural arragement of the protein is given, this is not mentioned for cathelicidins. Moreover, the specific described antimicrobial activity should be added in the style of line 86-88 for cathelicidins.  

Chapter 2.2. are re-organized to follow the same pattern as chapter 2.1 as suggested by the reviewer.

Both terms ‚antimicrobial peptide‘ and ‚host defence peptide‘ are used as synonyms. While only the term AMP was used in the introductory part, this changed to HDP in chapter 2. It would be good if the authors used one term consitently or added that they can be used synonymously. Possibly it is meant differently.

This point was raised by both reviewers, thus it has been clarified in the abstract that HDPs also often are called AMPs. And the term HDP are more consistently used throughout the manuscript.

For chapter 4 the heading is missing (line 150) or wrongly numbered.

The heading was mislabeled and has been corrected in the revised manuscript (line197).

Line 132-133 should read epidermis.

This typo and other typographical errors have been corrected.

It seems a word is missing in the sentence beginning line 256. Please check.

This sentence has been rephrased, now reading: “Nanoparticles lipid carriers (NLCs) encapsulating LL-37 and administered through the topical route, have demonstrated to accelerated wound closure, re-epithelization and decreased the inflammation in vitro and in vivo [130].” (Line 296-298)

Occasionally, some minor spelling mistakes occured throughout the manuscript, i.e. line 177 -179. Please recheck.

Some miss spellings and typos have been corrected and marked throughout the revised manuscript.

The list of references is not consistently formatted in the same style. Please check especially ref.4, 5 and 19-21. In some references the doi is given plus page numbers and volume and in others it is not or only the doi.  

The reference list has been reformatted, and as the reviewer also indicate that the references not are adequate, we have also added 9 additional references in addition to the 4 references suggested by reviewer #2.

Reviewer 2 Report

The submitted manuscript “Immunomodulatory properties of host defense peptides in skin wound healing” by Petkovic et al summarised the recent advances of host defense peptide/antimicrobial peptides related molecular mechanism and their immune-modulatory features. The contents were sufficiently substantial and broad-ranging to allow coverage of the field. It is a clear and well-organised mini-review. Therefore, I recommend this manuscript be published after minor revision.

There are few minor comments,

  1. page 1, line 40, they briefly mentioned the host defense properties of AMPs. However, they should also refer to the recent excellent rev (such as Lancet Infect Dis 2020; 20: e216–30, https://doi.org/10.1016/S1473-3099(20)30327-3))
  2. page 2 line 48, they described the AMP and peptidomimetics. But a recent comprehensive rev on AMP and peptidomimetics should be also referred (Chem. Soc. Rev., 2021,50, 4932-4973 https://doi.org/10.1039/D0CS01026J)
  3. page 2 line 60, the authors provided the introduction of AMPs in wound healing. However, two refs on AMPs in wound healing should be briefly described. (https://molmed.biomedcentral.com/articles/10.2119/2008-00002.Steinstraesser; https://www.tandfonline.com/doi/full/10.1586/14787210.2015.1033402)
  4. figure 1, they should define the abbreviations, including EGF, VEGF…, in the caption.
  5. Page 3, line 76, there is no definition of “HDs”.
  6. Page 7, line 215, there is no definition of “ERK” and “Akt”
  7. In the whole text, the authors should be consistent with the usage of “Antimicrobial peptides”, “AMPs”, “host defense peptides”, “HDPs”.

Author Response

Dear Reviewer

Thanks for a thorough and constructive review.

All points have been addressed and marked in the revised manuscript.

The points raised are marked in italic with a response following:

  1. page 1, line 40, they briefly mentioned the host defense properties of AMPs. However, they should also refer to the recent excellent rev (such as Lancet Infect Dis 2020; 20: e216–30, https://doi.org/10.1016/S1473-3099(20)30327-3))
  2. page 2 line 48, they described the AMP and peptidomimetics. But a recent comprehensive rev on AMP and peptidomimetics should be also referred (Chem. Soc. Rev., 2021,50, 4932-4973 https://doi.org/10.1039/D0CS01026J)
  3. page 2 line 60, the authors provided the introduction of AMPs in wound healing. However, two refs on AMPs in wound healing should be briefly described. (https://molmed.biomedcentral.com/articles/10.2119/2008-00002.Steinstraesser; https://www.tandfonline.com/doi/full/10.1586/14787210.2015.1033402)

All four references have been introduced and added in the revised manuscript, line 47, 62, 87, and 77, respectively.

  1. figure 1, they should define the abbreviations, including EGF, VEGF…, in the caption.

The abbreviations have been written out line 95.

  1. Page 3, line 76, there is no definition of “HDs”.

 The abbreviations is defined on line 102.

  1. Page 7, line 215, there is no definition of “ERK” and “Akt”

 The abbreviations have been written out line 265.

  1. In the whole text, the authors should be consistent with the usage of “Antimicrobial peptides”, “AMPs”, “host defense peptides”, “HDPs”.

It has been clarified in the abstract that HDPs also often are called AMPs. And the term HDP are more consistently used throughout the manuscript.